# Cell elongation is regulated through a central circuit of interacting transcription factors in the Arabidopsis hypocotyl

**Eunkyoo Oh, Jia-Ying Zhu, Ming-Yi Bai, Rafael Augusto Arenhart, Yu Sun, Zhi-Yong Wang\***

Department of Plant Biology, Carnegie Institution for Science, Stanford, United States

**Abstract** As the major mechanism of plant growth and morphogenesis, cell elongation is controlled by many hormonal and environmental signals. How these signals are coordinated at the molecular level to ensure coherent cellular responses remains unclear. In this study, we illustrate a molecular circuit that integrates all major growth-regulating signals, including auxin, brassinosteroid, gibberellin, light, and temperature. Analyses of genome-wide targets, genetic and biochemical interactions demonstrate that the auxin-response factor ARF6, the light/temperature-regulated transcription factor PIF4, and the brassinosteroid-signaling transcription factor BZR1, interact with each other and cooperatively regulate large numbers of common target genes, but their DNA-binding activities are blocked by the gibberellin-inactivated repressor RGA. In addition, a tripartite HLH/bHLH module feedback regulates PIFs and additional bHLH factors that interact with ARF6, and thereby modulates auxin sensitivity according to developmental and environmental cues. Our results demonstrate a central growth-regulation circuit that integrates hormonal, environmental, and developmental controls of cell elongation in Arabidopsis hypocotyl.

**\*For correspondence:**
zywang24@stanford.edu

**Competing interests:** The authors declare that no competing interests exist.

**Reviewing editor**: Sheila McCormick, University of California-Berkeley & USDA Agricultural Research Service, United States

## Introduction

The high levels of developmental plasticity in higher plants relies on coordinated regulation of cell elongation by many hormonal and environmental signals, including particularly light, temperature, auxin, gibberellin (GA), and brassinosteroid (BR), which have major effects on cell elongation and seedling morphogenesis. Complex interplays among these signals have been observed at the genetic and physiological levels, but the molecular mechanisms underlying these interactions are not fully understood. Recent studies have shown integration of the light, temperature, BR and GA pathways through direct interactions between their target transcription regulators (*Gallego-Bartolome et al., 2012*; *Oh et al., 2012*; *Bai et al., 2012b*; *Li et al., 2012b*). However, the relationship of the major growth hormone auxin with the other signals remains unclear at the molecular level.

Auxin is the dominant plant growth hormone that plays key roles in nearly all developmental processes including patterning and growth responses to the environment. Auxin is essential for cell elongation responses to shade, warm temperature, and the circadian clock as well as tropic growth responses to light and gravity (*Stewart and Nemhauser, 2010*; *Del Bianco and Kepinski, 2011*). While regulation of auxin level and distribution is an important aspect of its function, the ability of auxin to regulate cell elongation also depends on developmental context and the status of other hormonal and environmental signals (*Stewart and Nemhauser, 2010*; *Del Bianco and Kepinski, 2011*). For example, auxin and brassinosteroid (BR) are known to be interdependent and show synergistic interactions in promoting hypocotyl elongation, and they induce highly overlapping transcriptional responses (*Goda et al., 2004*; *Nemhauser et al., 2004*). Normal auxin response also requires another growth-promoting hormone gibberellin (GA) (*Chapman et al., 2012*). Furthermore, both auxin level

**eLife digest** Plants can grow by making more cells or by increasing the size of these existing cells. Plant growth is carefully controlled, but it must be able to respond to changes in the plant's environment.

Many different plant hormones and various signals from the environment—such as light and temperature—influence how and when a plant grows. The different signals that affect cell growth typically act via distinct pathways that change which genes are switched on or off inside the cells. However, the ways in which these different signals are coordinated by plants are not fully understood.

Now, Oh et al. have looked at the genes that are switched on and off in response to all the major signals that regulate the growth of the first stem to emerge from the seed of *Arabidopsis*, a small flowering plant that is widely studied by plant biologists. Oh et al. found that the proteins that change gene expression in response to hormones or the environment bind to each other. These proteins, which are collectively called transcription factors, were also revealed to cooperate to regulate the expression of hundreds of genes: transcription factors have not been seen to behave in this way in plants before.

By discovering a central mechanism that coordinates the different signals that control plant growth, these findings may guide future efforts to boost the yields of food crops and plants that are grown to make biofuels.

and sensitivity are modulated by light, temperature, and circadian rhythm, through the phytochrome-interacting factor (PIFs) family of bHLH transcription factors (*Covington and Harmer, 2007*; *Nozue et al., 2007*; *Koini et al., 2009*; *Franklin et al., 2011*; *Nozue et al., 2011*; *Hornitschek et al., 2012*; *Sun et al., 2012*; *Li et al., 2012a*). The mechanisms for regulation of auxin levels and distribution through metabolism and polar auxin transport have been studied extensively, however, little is known about direct interaction between the signal transduction pathways of auxin and other signals at the molecular level.

Auxin signaling induces ubiquitination and degradation of the AUX/IAA family proteins to release their inhibition of the auxin response factor (ARF) family of transcription factors (*Chapman and Estelle, 2009*; *Vernoux et al., 2011*), but target genes of ARFs remain largely unknown and thus the mechanisms linking ARF activation to context-specific cellular responses remain poorly understood (*Del Bianco and Kepinski, 2011*). By contrast, BR acts through a receptor kinase pathway to inhibit BIN2/GSK3-mediated phosphorylation of the brassinazole resistant (BZR) family of transcription factors, leading to their accumulation in the nucleus and regulation of thousands of target genes (*Wang et al., 2012*). Cross regulation between auxin and BR has been observed at several levels, including auxin activation of BR biosynthetic genes, BR regulation of the expression levels of auxin transporters (*Bao et al., 2004*; *Mouchel et al., 2006*; *Chung et al., 2011*; *Yoshimitsu et al., 2011*), and BIN2 phosphorylation of ARF2 (*Vert et al., 2008*). However, these cross regulation mechanisms appear insufficient to explain the mutual interdependence between BR and auxin. There has been evidence that both BZR2 (also named BES1) and ARF5 bind to the promoter of the auxin- and BR-activated *SAUR15* gene (*Walcher and Nemhauser, 2012*). The functions of such interactions in the auxin-BR co-regulation of genome expression and cell elongation, however, remain unclear. Much less is known about direct interactions between auxin and the phytochrome or GA pathways. While recent studies demonstrated convergence of the BR, light, and GA pathways through interactions between PIF4, BZR1 and the GA-inactivated repressor DELLA proteins (*Feng et al., 2008*; *de Lucas et al., 2008*; *Gallego-Bartolome et al., 2012*; *Oh et al., 2012*; *Wang et al., 2012*; *Bai et al., 2012b*; *Li et al., 2012b*), the current models suggest that auxin interacts with other signals mainly through modulation of hormone levels (*Mouchel et al., 2006*; *Chung et al., 2011*; *Franklin et al., 2011*; *Yoshimitsu et al., 2011*; *Chapman et al., 2012*; *Sun et al., 2012*; *Li et al., 2012a*).

In this study, we performed genome-wide analyses of target genes of an auxin response factor (ARF6) that regulates hypocotyl elongation, and we demonstrate that the majority of ARF6 target genes are also targets of BZR1 and/or PIF4. Genetic and biochemical assays further demonstrate that these factors interact directly and bind to shared target genes cooperatively. Furthermore, the DELLA

protein RGA interacts with ARF6 and blocks its DNA binding. Our study elucidates a central growth regulation circuit that explains how auxin, BR, GA, light, and temperature act together in regulating hypocotyl cell elongation and how the hormone sensitivities are modulated by environmental signals and developmental programs.

## Results

### ARF6 shares genomic targets with BZR1 and PIF4

ARF6 and its closed homolog ARF8 were previously shown to redundantly regulate hypocotyl elongation in *Arabidopsis* (*Nagpal et al., 2005*). To define the genomic targets of auxin involved in hypocotyl cell elongation, we performed chromatin-immunoprecipitation followed by sequencing (ChIP-Seq) analysis of target genes of ARF6. An ARF6-Myc fusion protein was expressed from the *ARF6* promoter in transgenic *arf6-2;arf8-3* plants, and rescued the short-hypocotyl phenotype of the *arf6;arf8* double mutant (*Figure 1—figure supplement 1A*). ChIP-Seq analysis using anti-Myc antibody identified 2037 ARF6-binding sites in the *Arabidopsis* genome. Most of the ARF6 binding sites were in the gene promoter regions consistent with its molecular function as a transcription regulator (*Figure 1A*). The 2037 binding sites were linked to 2675 neighbor genes (*Figure 1— source data 1A*), which were considered ARF6 binding target genes. The ARF6 binding targets include 40 of the 49 early auxin-induced genes in the hypocotyl tissues after 30 min of auxin treatment (*Chapman et al., 2012*), but only 1 of the 16 immediate auxin-repressed genes (*Figure 1B*). Therefore, ARF6 appears to function mainly as a transcriptional activator, consistent with previous study (*Tiwari et al., 2003*). Comparison with auxin-activated genes (*Chapman et al., 2012*) identified 255 ARF6 binding targets that are activated by auxin in the hypocotyl tissues. These include many genes known to promote cell elongation (*PREs*, *BIM1*, *BEE1* and *HAT2*, *SAURs*) and many genes with known function in auxin response, such as *AUX/IAAs*, *PINs* and *PINOID* (*Figure 1— source data 2*).

To understand the relationships between auxin, BR, and phytochrome pathways in regulating genome expression, we compared the ARF6 targets with the genome targets of BZR1 and PIF4 (*Figure 1—source data 1*). To generate comparable data sets, we performed BZR1 ChIP-Seq analysis using dark-grown seedlings (*Figure 1—source data 1B*), as used in the ARF6 and PIF4 ChIP-Seq (*Oh et al., 2012*). Interestingly, large portions of ARF6 binding targets overlapped with either targets of BZR1 (51%) or PIF4 (71%) or both (42%) (*Figure 1C*). The common targets include many genes with known functions in cell elongation (*EXP8*, *BIM1*, *BEE1/3*, *PREs*, *HAT2*, *IBH1*, *HFR1*, *PAR1/2*, *EXO*) and auxin response (*PINs* and *SAURs*). Moreover, the binding peak patterns of ARF6, BZR1, and PIF4 seemed very similar on promoters of many common targets (*Figure 1—figure supplement 1B*), and the overall binding peaks were very close to each other (*Figure 1—figure supplement 1C*), indicating that these three transcription factors bind to same or nearby genomic locations. ChIP-reChIP assays showed that two common targets, *SAUR15* and *IAA19*, were recovered by sequential immunoprecipitation in plants expressing both BZR1-YFP and ARF6-Myc, but not in plants expressing ARF6-Myc only (*Figure 1D*), indicating that ARF6 and BZR1 co-occupy these promoters.

Cis-element analysis identified two types of E-box motifs, the G-box (CACGTG), and the Hormone Up at Dawn (HUD, CACATG) motifs (*Michael et al., 2008*), that are highly enriched in the ARF6 binding regions associated with the auxin-activated genes (*Figure 1E*). These motifs are known binding sites for BZR1/2 and PIFs and are also over represented in the genomic regions bound by BZR1 and PIF4 (*Sun et al., 2010*; *Oh et al., 2012*). The canonical ARF binding motif, AuxRE (TGTCTC), and the binding motif (TGTCGG) recently identified for recombinant ARF1 and ARF5 were also highly enriched among the ARF6 binding regions (*Figure 1E*; *Ulmasov et al., 1999*; *Boer et al., 2014*). About half of the auxin-activated ARF6 binding regions had both core AuxRE (TGTC) and the E-box motifs within the ±100 base pairs around the binding site, which is much higher than random expectation (only 9%) (*Figure 1F*). The E-box motifs tend to be located close to the core AuxRE, mostly within 20 base pairs (*Figure 1G*). Furthermore, the ARF6 binding regions having both AuxRE and E-box motifs were significantly more frequently associated with auxin-activated genes than binding regions having only AuxRE (*Figure 1H*). Consistent with cis-elements clustering, the ARF6 targets shared by BZR1 and PIF4 had higher percentage of auxin-activated genes than the targets of ARF6 alone, and a higher percentage of BR-activated genes than genes that are targets of BZR1 only (*Figure 1I*). Furthermore, the GA-activated genes and light-repressed genes were also highly enriched

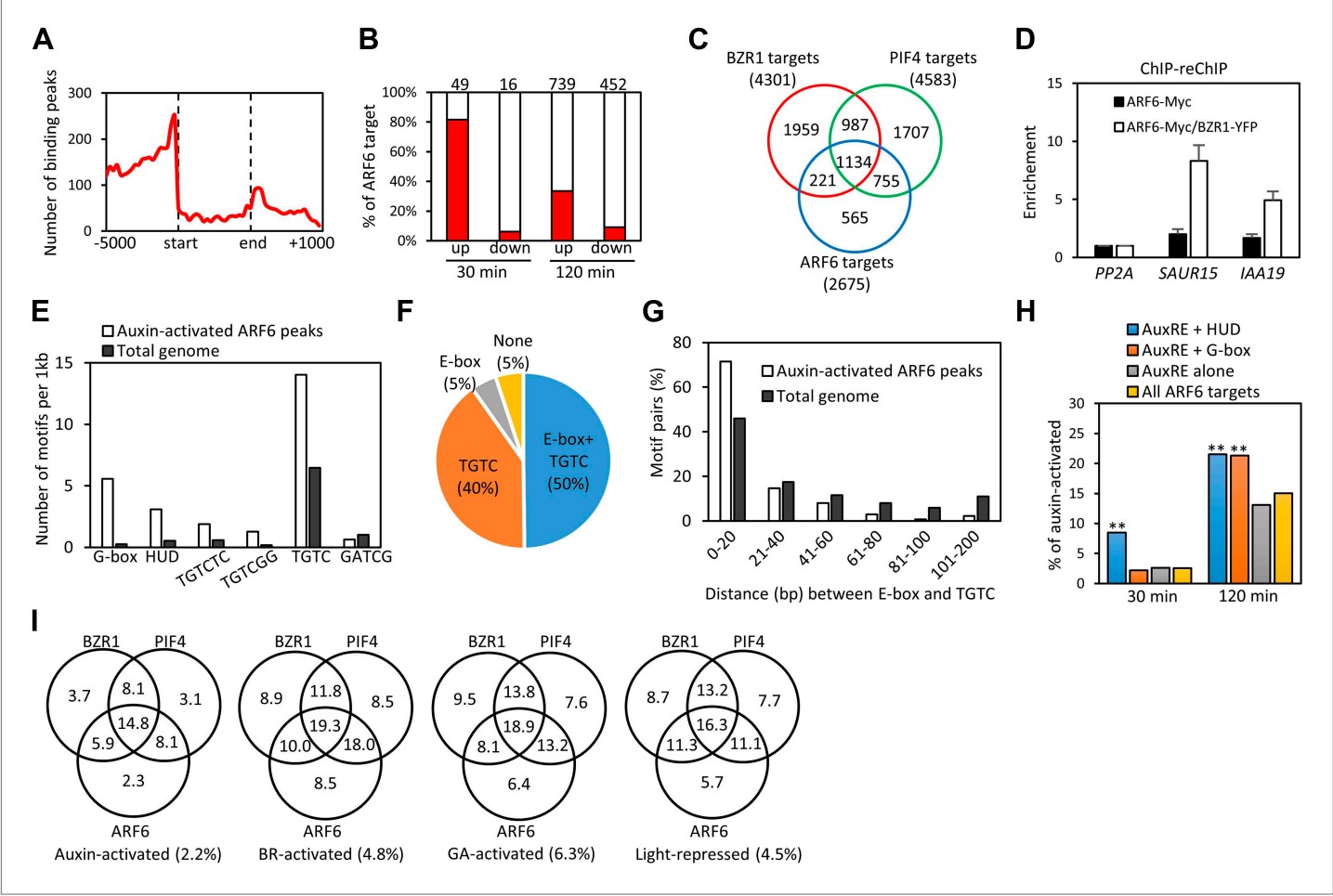

**Figure 1**. ARF6 ChIP-Seq analyses. (**A**) Distribution of ARF6 binding peaks relative to gene structure (−5000 base pairs from transcription start site to +1000 base pairs downstream of 3′ end). (**B**) Most of the early auxin-activated genes are ARF6 targets. Numbers above the columns indicate number of genes up- or down-regulated by 30 or 120 min of auxin treatments. (**C**) Venn diagram shows significant overlap among binding target genes of BZR1, PIF4 and ARF6. (**D**) ChIP-reChIP assay shows that BZR1 and ARF6 co-occupy shared target promoters. The enrichment of precipitated DNA was calculated as the ratio between transgenic plants and wild type control, normalized to that of the *PP2A* coding region as an internal control. Error bars indicate the SD of three biological repeats. (**E**) The G-box (CACGTG), HUD (CACATG), canonical AuxRE (TGTCTC) and TGTCGG are enriched in the ARF6 binding peaks associated with auxin-activated genes. GATCG (a random motif) is shown as a negative control. (**F**) Percentages of auxin-activated ARF6 binding peaks that have both E-box motif and core AuxRE (TGTC), only TGTC, or only E-box motifs. (**G**) Distribution of distance between E-box motifs and core AuxRE (TGTC) found in the ARF6 peaks associated with auxin-activated genes or total *Arabidopsis* genome. (**H**) ARF6 binding peaks having both E-box motifs and AuxRE have higher probability (%) of being associated with auxin-activated (30 or 120 min treatment) genes than the ARF6 binding peaks having only AuxRE. **$p<0.01$. (**I**) Venn diagram shows that genes activated by auxin, BR, or GA and genes repressed by light are enriched in the common binding targets of BZR1, PIF4 and ARF6. Numbers in the Venn diagram indicate percentage of corresponding genes (e.g., auxin-activated genes) in each section. Numbers in parentheses indicate percentage of genes in total *Arabidopsis* genome.

The following source data and figure supplements are available for figure 1:

**Source data 1**.
**Source data 2**. Auxin-activated genes previously identified in hypocotyls (*Chapman et al., 2012*) were compared with ARF6 target genes identified by ChIP-Seq to identify the auxin-activated ARF6 target genes in hypocotyls.
**Figure supplement 1**.

among the common targets of BZR1, PIF4 and ARF6 (*Figure 1I*). These results suggest that the major growth signals—auxin, BR, GA, and light—converge at shared genomic target promoters containing combinatorial cis-elements for these factors.

## BZR1 and PIF4 interact with ARF6

The large number of common target genes of ARF6, BZR1 and PIF4 raises a possibility of direct interactions among these transcription factors. Indeed, ARF6 directly interacted with both BZR1 and PIF4 through the C-terminal domain of BZR1 and the bHLH domain of PIF4 in the yeast two-hybrid assays (*Figure 2A–D*). Both the middle and C-terminal domains of ARF6 were required for the interactions with BZR1 and PIF4 (*Figure 2—figure supplement 1*). Co-immunoprecipitation assays showed that ARF6 interacts with BZR1 and PIF4 in vivo (*Figure 2E*). In addition, the ARF6–PIF4 interaction was increased by co-transfection with a gain-of-function bzr1-1D protein that is constitutively active irrespective of BR signaling (*Wang et al., 2002*; *Figure 2F*), suggesting that BZR1–PIF4 interaction enhances PIF4 interaction with ARF6. We next examined if BZR1 and PIF4 interact with other ARFs using yeast two-hybrid assays, and the results show that both BZR1 and PIF4 specifically interacted with ARF8, but not ARF1 and ARF7 (*Figure 2G,H*, *Figure 2—figure supplement 2*), suggesting that BZR1 and PIF4 mediate only subsets of auxin responses such as hypocotyl elongation by interacting with specific ARFs.

To test if ARF6 forms cooperative DNA binding complexes with BZR1 and PIF4, we measured in vivo ARF6 DNA-binding activity on the ARF6, BZR1, and PIF4 common targets. The ARF6 binding to the promoters of common target genes were increased by BR treatment, as observed for ARF5 on the promoter of *SAUR15* (*Walcher and Nemhauser, 2012*; *Figure 2—figure supplement 3*). In addition, *bzr1-1D* and *PIF4-OX* enhanced ARF6 binding to the common targets (*SAUR15*, *SAUR19* and *At1g29500*), but not to the targets of ARF6 only (*At4g12110* and *At2g40880*), indicating that BZR1 and PIF4 enhance ARF6 DNA-binding in vivo (*Figure 2I,J*). Based on the signal intensity of ARF6 ChIP-Seq data, ARF6 occupancy is higher at binding regions containing both E-box motifs and AuxRE than regions containing only AuxRE (*Figure 2K*). These results support cooperative DNA binding of BZR1/PIF4 and ARF6.

## ARF6, BZR1, and PIF4 interdependently activate shared target genes

To evaluate the function of BZR1-ARF6-PIF4 interaction in regulating gene expression, we carried out RNA-Seq analyses using wild type and the *iaa3/shy2-2* mutant seedlings treated with mock or BL for 4 hr. The gain-of-function *iaa3/shy2-2* mutation causes auxin insensitivity by stabilizing the IAA3 protein, which interacts with and inactivates multiple ARFs including ARF6 and ARF8 (*Figure 3—figure supplement 1*; *Vernoux et al., 2011*). We identified 2664 genes that responded to BR in wild-type plants and 4725 genes differentially expressed in the *iaa3* mutant compared to wild type (*Figure 3A*, *Figure 3—source datas 1 and 2*). Expression levels of many BR-regulated genes (1465, 55%) were also affected by *iaa3*, and mostly in opposite ways (correlation coefficient = −0.6) (*Figure 3B*). Of 2482 BZR1- and PIFs-co-regulated genes (*Oh et al., 2012*), 976 genes were affected by *iaa3* (350 expected randomly) (*Figure 3C*). Heat-map in *Figure 3D* shows that most of the co-regulated genes (70%) are similarly regulated by BZR1 and PIFs, but oppositely affected by *iaa3*. Gene ontology (GO) analysis showed that many auxin-responsive genes and genes involved in cell wall biogenesis are activated by BZR1 and PIFs but repressed by *iaa3* (*Figure 3—figure supplement 2A*).

The overall effect of BR treatment on gene expression is diminished in *iaa3* compared to wild type (*Figure 3E*). Of the 1616 BR-activated genes detected in wild-type plants, only 276 genes (17.1%) were activated by BR treatment in *iaa3*, whereas a bigger portion of BR-repressed genes (30% of 1048) were still repressed by BR treatment in *iaa3* (*Figure 3F*, *Figure 3—source data 1*). Reverse transcription-quantitative PCR (RT-qPCR) analysis of selected genes, including *SAURs* (*SAUR15, SAUR26*), *ESPANSIN* (*EXP8*), *PREs*, and *DWF4,* confirmed the patterns observed in the genome-wide analysis (*Figure 3G*). Activation of the BR- and auxin-induced genes by *bzr1-1D* was abolished or diminished by *iaa3*, but the repression of BR-repressed *DWF4* expression was unaffected by *iaa3* (*Figure 3G*). Consistent with *iaa3* mutation not affecting the BZR1 function in repression of gene expression, the phosphorylation or accumulation status of BZR1 was not affected in the *iaa3* mutant (*Figure 3—figure supplement 2B,C*). If the effect of *iaa3* on BR responses is due to inactivation of ARFs, loss of the ARF functions would have a similar effect as *iaa3*. Indeed, BR-activated genes were less activated in *arf6;arf8* than in wild type, but BR-repressed genes were normally repressed in *arf6;arf8* (*Figure 3H*). The effect of *arf6;arf8* is similar to but weaker than *iaa3* (*Figure 3G,H*), consistent with additional ARF factors playing overlapping role with ARF6 and ARF8 and being suppressed by *iaa3*. These results indicate that BR activation of genes for hypocotyl elongation is dependent on auxin activation of ARFs, whereas BR feedback repression of BR biosynthesis genes is independent of auxin signaling.

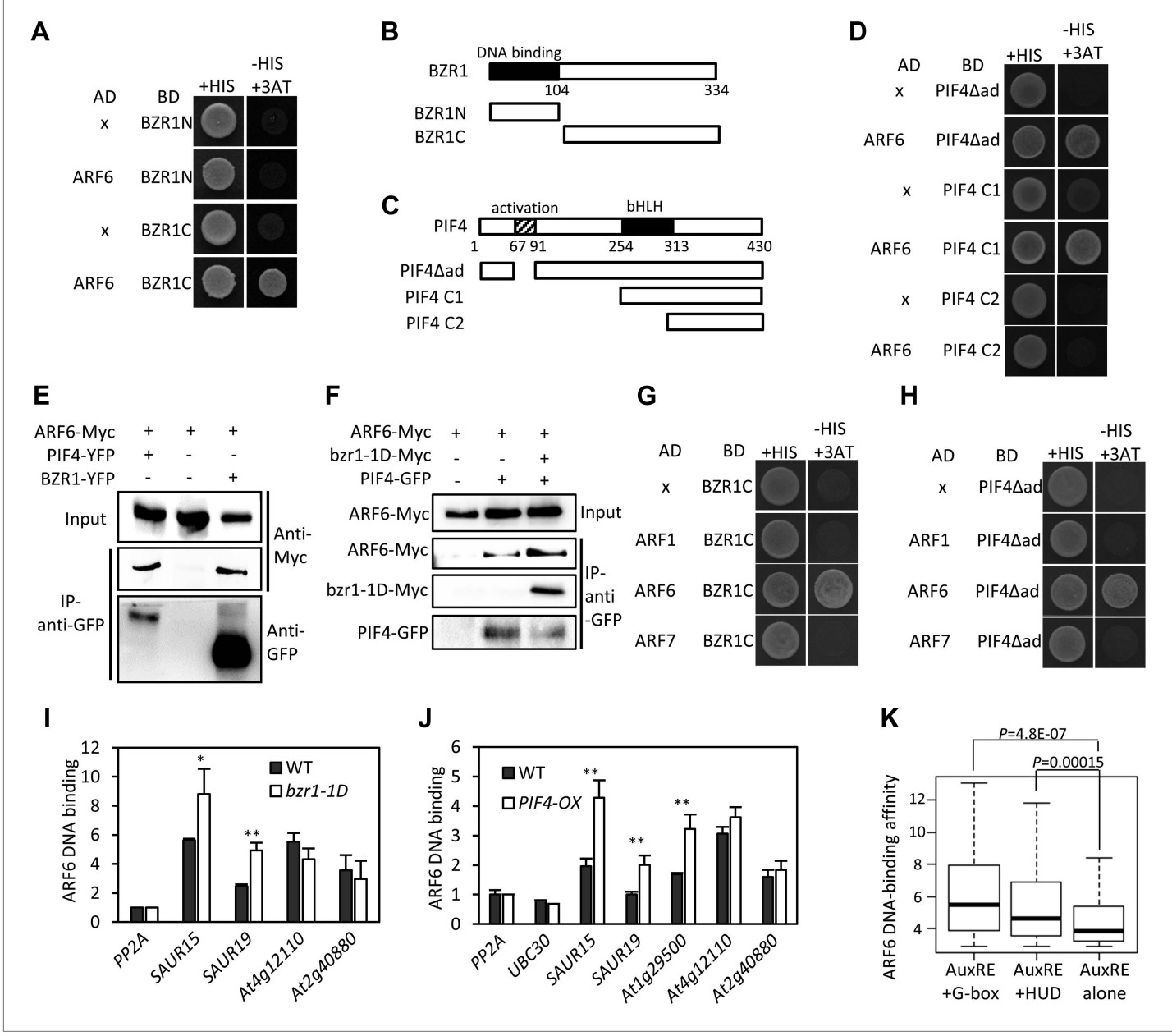

**Figure 2**. ARF6 interacts with BZR1 and PIF4. (**A**) ARF6 interacts with BZR1 in yeast two-hybrid assay. Yeast clones were grown on the synthetic dropout (+HIS) medium or synthetic dropout medium without histidine (−HIS) plus 1 mM 3AT. (**B** and **C**) Box diagram of various fragments of BZR and PIF4 used in (**A**, **D**, **G**, **H**). (**D**) ARF6 interacts with PIF4 in yeast two-hybrid assay. (**E**) ARF6 interacts with BZR1 and PIF4 in vivo. Transgenic plants expressing the indicated fusion proteins were used for immunoprecipitation using anti-GFP antibody, and the immunoblots were proved with anti-Myc antibody to detect interaction with the Myc-tagged ARF6 protein. (**F**) BZR1 enhances the ARF6–PIF4 interaction. Arabidopsis mesophyll protoplasts were transfected to express ARF6-Myc alone or together with PIF4-GFP and bzr1-1D-Myc as indicated, and the extracted proteins were immunoprecipitated by anti-GFP antibody. Gel blots were probed with anti-Myc or anti-GFP antibody. (**G** and **H**) BZR1 (**G**) and PIF4 (**H**) interact with ARF6, but not with ARF1 and ARF7 in yeast two-hybrid assays. (**I**) ARF6 DNA-binding is enhanced by *bzr1-1D*. Seedlings (*35S::ARF6-Myc* (WT) and *35S::ARF6-Myc;bzr1-1D* (*bzr1-1D*)) grown on the 2 μM PPZ in the dark for 6 days were used for ChIP assays. Error bars in the (**I** and **J**) indicate the SD of three biological repeats. *p<0.05 and **p<0.01. (**J**) *PIF4-OX* enhances ARF6 DNA-binding. Seedlings (*35S::ARF6-Myc* (WT) and *35S::ARF6-Myc;PIF4-OX* (*PIF4-OX*)) grown under light were used for ChIP assays of ARF6 binding to the indicated promoters. (**K**) Box plot shows that ARF6 binding peaks having both E-box motifs and AuxRE tend to have higher ARF6 DNA-binding affinity. ARF6 DNA-binding affinity was based on the peak score from the ARF6 ChIP-Seq analysis with CSAR.

*Figure 2. Continued on next page*

*Figure 2. Continued*

The following figure supplements are available for figure 2:

**Figure supplement 1**.

**Figure supplement 2**. ARF8 interacts with both BZR1 and PIF4.

**Figure supplement 3**. ARF6 DNA-binding on the common targets of ARF6 and BZR1 is enhanced by BR treatment.

We next asked whether BZR1 and PIFs are involved in the auxin regulation of gene expression. Auxin-activated genes *SAUR15*, *SAUR19*, *ACS5*, and *At4g13790* were less activated by auxin in *bri1-116* than in wild type, but *bzr1-1D* enhanced the auxin activation of these genes in the *bri1-116* background (*Figure 3I*), indicating that BR promotes auxin responsive genes by activating BZR1. The auxin activation of these genes was reduced in the *pif-quadruple* mutant (*pifq*) lacking four PIFs (PIF1/PIL5, PIF3, PIF4 and PIF5/PIL6) (*Shin et al., 2009*) and completely abolished by overexpression of PAR1 (*PAR1-OX*), which inhibits PIF activities (*Hao et al., 2012*; *Figure 3I*), consistent with previous observation (*Roig-Villanova et al., 2007*). Taken together, our genome- and gene-expression analyses show that BZR1, PIFs, and ARFs interdependently regulate the expression of large numbers of genes, integrating BR, light, and auxin signals into a common set of transcriptome.

## ARF6, BZR1, and PIF4 synergistically promote hypocotyl elongation

The BZR-ARF-PIF module provides a molecular model for integrating auxin signaling with BR and phytochrome pathways. To understand the functional importance of the interactions between ARF6, BZR1, and PIF4, we analyzed the effects of genetic alteration of each component on the growth responses to changes in the other activities. Hypocotyl elongation of the BR receptor mutant *bri1* shows diminished response to auxin (*Figure 4A*), as observed previously (*Nemhauser et al., 2004*), but the auxin-insensitive phenotype of *bri1* was fully rescued by *bzr1-1D* (*Figure 4A*), indicating that BZR1 mediates BR enhancement of auxin response. The hypersensitivity of *bzr1-1D* to auxin was abolished by the *iaa3* mutation (*Figure 4B*), suggesting that ARF activity is required for BZR1 function. Consistently, both *iaa3* and *arf6;arf8* were less sensitive to BR than was wild type (*Figure 4C*). Furthermore, the *bzr1-1D;arf6;arf8* triple mutant and *bzr1-1D;iaa3* double mutant showed shorter hypocotyls on the medium containing BR biosynthesis inhibitor BRZ than the *bzr1-1D* single mutant (*Figure 4D,E*), indicating that ARF6/8 are required for BZR1 promotion of hypocotyl elongation. Finally, to determine whether PIFs are required for auxin response, we checked the hypocotyl response to auxin in *pifq* and *PAR1-OX*. Compared with wild type, *pifq* was less sensitive and *PAR1-OX* was almost insensitive to auxin (*Figure 4A*). These results indicate that ARF, BZR1 and PIFs are interdependent in promoting hypocotyl elongation, consistent with their cooperative regulation of a core set of genes involved in hypocotyl cell elongation.

## The HLH/bHLH module mediates developmental regulation of auxin sensitivity

The stronger auxin insensitivity of *PAR1-OX* than *pifq* suggests that besides the four PIFs, additional PAR1-inactivated factors are involved in the auxin signaling. However, PAR1 did not interact with IAA3 and ARF6 (*Figure 5—figure supplement 1*). The *Arabidopsis* Interactome map showed that PAR1 interacts with another bHLH transcription factor BEE2 (*Arabidopsis Interactome Mapping Consortium, 2011*), which was shown to be involved in BR regulation of cell elongation (*Friedrichsen et al., 2002*). Our yeast two-hybrid assays confirmed that PAR1 interacts with BEE2 and its close homolog HBI1 as well (*Figure 5A*). Like PIF4, BEE2, and HBI1 also interact with ARF6 (*Figure 5B*). Transgenic plants overexpressing a dominant repressor version of HBI1 (*HBI1-SRDX*) (*Bai et al., 2012a*) were less sensitive to auxin than was wild type (*Figure 5C*), supporting a positive role of HBI1 in auxin promotion of hypocotyl elongation. These results suggest that ARF6 interacts with multiple bHLH transcription factors as co-transcriptional regulators, and PAR1 attenuates auxin response through direct inactivation of these bHLH transcription factors.

HBI1 was recently identified as part of a tripartite HLH/bHLH cascade, in which two non-DNA binding HLH factors, PRE1 and IBH1, antagonize each other, and IBH1 interacts with HBI1 and inhibits

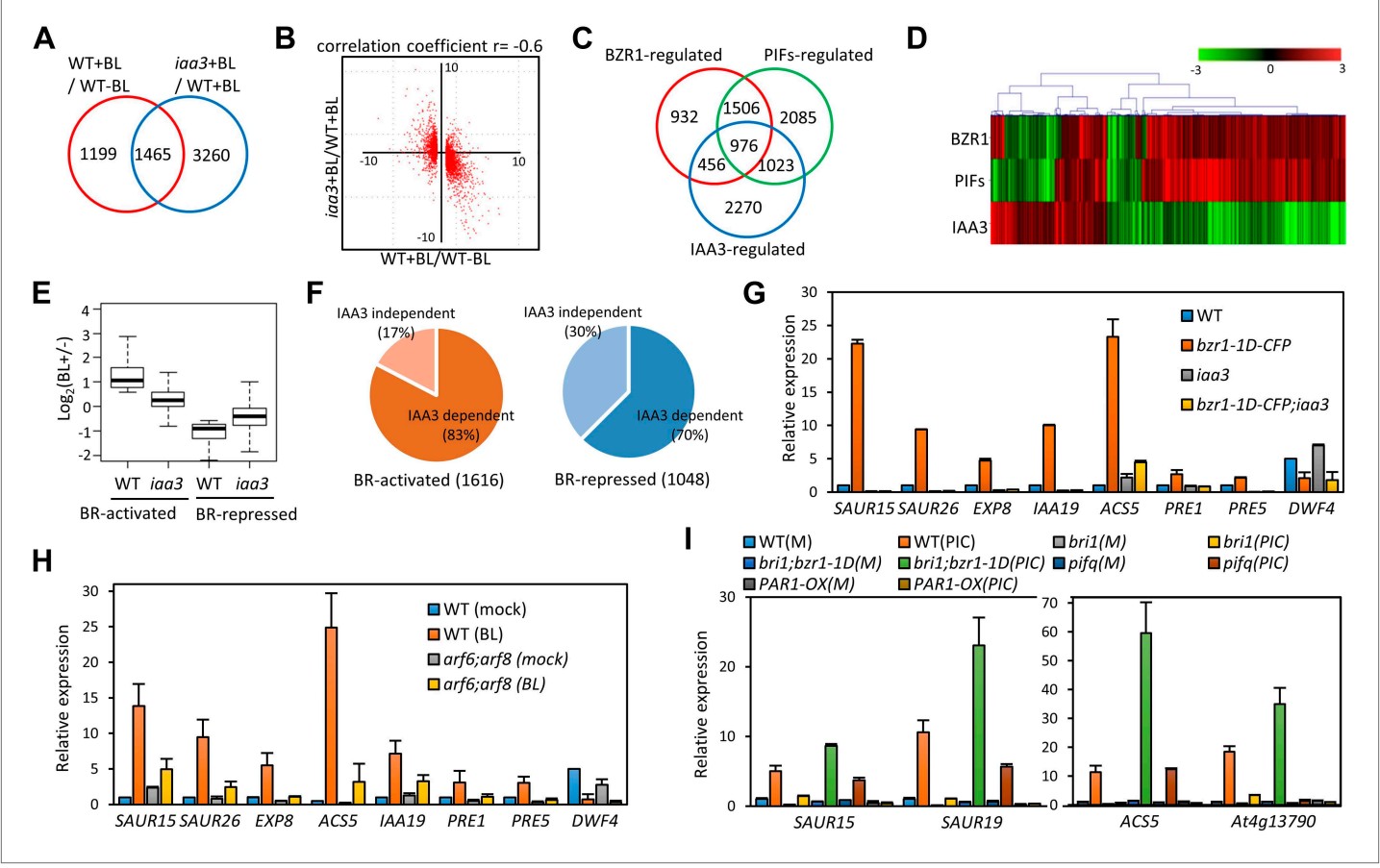

**Figure 3**. ARF6, BZR1, and PIF4 synergistically induce gene expression. (**A**) Significant overlap between BR-regulated genes and IAA3-regulated genes. (**B**) Scatter plot of log2-fold change values in the 1465 overlapping set of IAA3- and BR-regulated genes. (**C**) Significant overlap among BZR1-, PIFs-, and IAA3-regulated genes. (**D**) Heat map of the 976 genes co-regulated by BZR1, PIFs, and IAA3. Scale bar indicates fold changes (log2 value). (**E**) Box plot representation of the 1616 BR-activated or the 1048 BR-repressed genes in the WT and *iaa3/shy2-2*. (**F**) Percentage of IAA3-dependent and IAA3-independent BR-regulated genes. Genes that were not significantly affected by BR treatment in *iaa3/shy2-2* are defined as IAA3-dependent BR-regulated genes. (**G**) qRT-PCR analysis of BZR1-regulated genes in etiolated seedlings grown on 2 μM PPZ medium. Similar results are obtained from two independent biological repeats. Error bars indicate the SD of three technical repeats. (**H**) qRT-PCR analysis of BR-regulated genes in the seedlings treated with either mock or 100 nM BL for 4 hr. Error bars indicate the SD of three biological repeats. (**I**) qRT-PCR analysis of auxin responsive genes in the seedlings grown on medium containing no hormone (M) or 1 μM picloram, an artificial auxin. Error bars indicate the SD of three biological repeats.

The following source data and figure supplements are available for figure 3:

**Source data 1**. BR-regulated genes in wild type and their BR-responsive expression in the *iaa3* mutant.

**Source data 2**. Genes whose expression levels are affected in the *iaa3* mutant.

**Figure supplement 1**. IAA3 interacts with both ARF6 and ARF8.

**Figure supplement 2**.

HBI1 DNA-binding (***Bai et al., 2012a***). Therefore, PRE1 and IBH1 may modulate auxin response by altering HBI1 activity. Indeed, knock down of multiple *PREs* (*PRE1*, *PRE2*, *PRE5* and *PRE6*) by artificial micro-RNA (*pre-amiR*) or overexpression of IBH1 (*IBH1-OX*) (***Bai et al., 2012a, 2012b***), significantly reduced the sensitivity of transgenic plants to auxin in terms of both hypocotyl elongation and gene expression (***Figure 5C,D***).

The expression levels of *IBH1* and *PRE1* are developmentally regulated; *PRE1* expression is high in young and growing tissues, whereas *IBH1* expression is high in mature and growth-arrested tissues

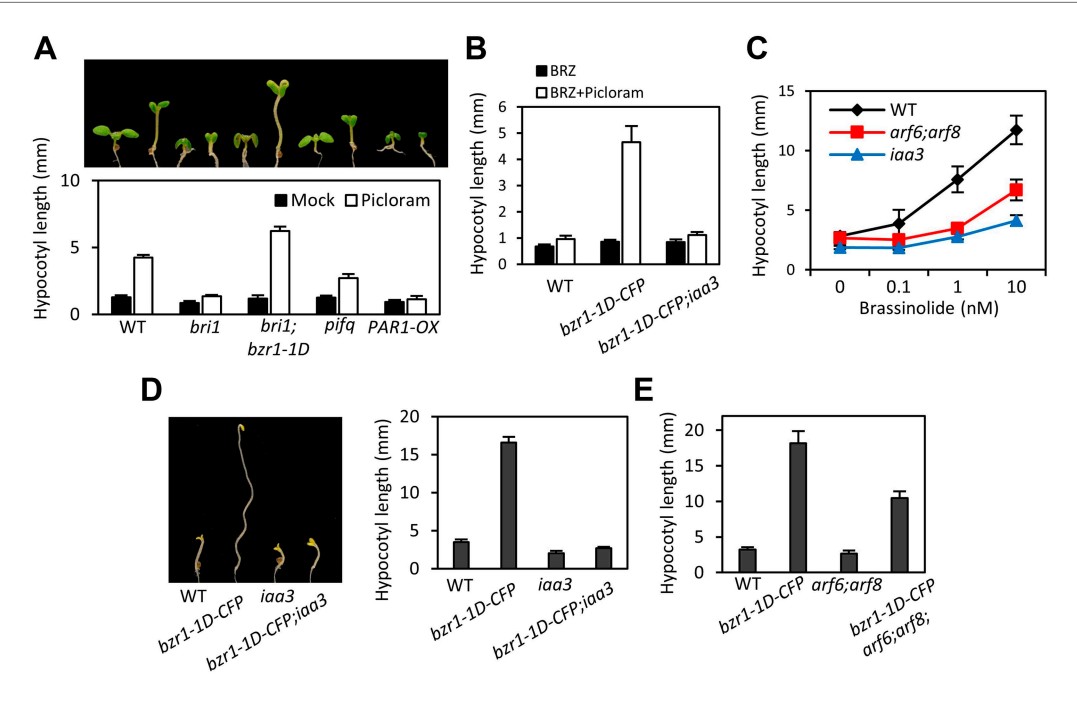

**Figure 4**. ARF6, BZR1, and PIF4 act interdependently in promoting hypocotyl elongation. (**A**) BZR1 and PIFs are required for auxin promotion of hypocotyl elongation. Seedlings were grown on 5 μM artificial auxin picloram or mock medium. (**B**) Hypersensitivity of *bzr1-1D* to auxin is abolished by *iaa3/shy2-2* mutation. Seedlings were grown on the medium containing 2 μM brassinazole (BRZ) with or without 5 μM artificial auxin picloram. (**C**) ARF6 and ARF8 are required for BR promotion of hypocotyl elongation. Seedlings were grown on medium containing 2 μM BRZ plus various concentration of BL in the dark. (**D**) The *iaa3/shy2-2* mutation inhibits BZR1 promotion of hypocotyl elongation. Representative seedlings are shown in left panel and quantification of hypocotyl lengths are shown in right graph. Seedlings were grown on the 2 μM BRZ medium in the dark. (**E**) ARF6 and ARF8 are required for BZR1 promotion of hypocotyl elongation. Seedlings were grown on the 2 μM BRZ medium in the dark. All error bars in (**A**–**E**) indicate SD (*n* = 10 plants).

(*Zhang et al., 2009*; *Ikeda et al., 2012*). We confirmed the developmental regulation of expression of *PRE1*, *PRE5* and *IBH1* (*Figure 5E*), and found that auxin treatment induced bigger fold changes of the *PREs* and *SAUR15* expression levels in young stems than in mature stems (*Figure 5E*). The reduced auxin sensitivity of the *SAUR15* gene in the mature stem was restored in the *PRE1-OX*, *HBI1-OX* and *PIF4-OX* plants (*Figure 5F*), supporting that developmental regulation of *PREs* and *IBH1* impacts auxin sensitivity through PIF4 and HBI1. The expression level of BR biosynthesis gene *DWF4* was high in young stems but low in mature stems (*Figure 5G*). Consistent with the expression pattern of *DWF4*, BZR1 was less phosphorylated in the young stems than in the mature stems (*Figure 5H*), indicating that BZR1 is more activated in the young tissues (*He et al., 2002*; *Gampala et al., 2007*; *Ryu et al., 2007*). Since BZR1 interacts with ARF6 to potentiate auxin response, differential BZR1 activity may also contribute to the difference in auxin sensitivity between young and mature tissues. Indeed, activation of BZR1 by the *bzr1-1D* mutation increased auxin sensitivity of *SAUR15* expression in the mature stem (*Figure 5I*). Together, these results suggest that developmental regulation of BZR1 and HLH factors contributes to the changes of auxin sensitivity over the progression of organ development.

## Auxin and GA crosstalk through RGA interaction with ARF6

GA regulates cell elongation through the degradation of DELLA proteins, which inactivate BZR1 and PIFs (*Bai et al., 2012b*; *de Lucas et al., 2008*; *Feng et al., 2008*). A comparison of ARF6 targets with BZR1 and PIF4 targets revealed that GA-activated genes are enriched in the common targets of ARF6, BZR1, and PIF4 (*Figure 1I*), suggesting that ARF6 is also involved in GA response. Therefore, we tested whether ARF6 directly interacts with the DELLA protein RGA. In yeast two-hybrid assays,

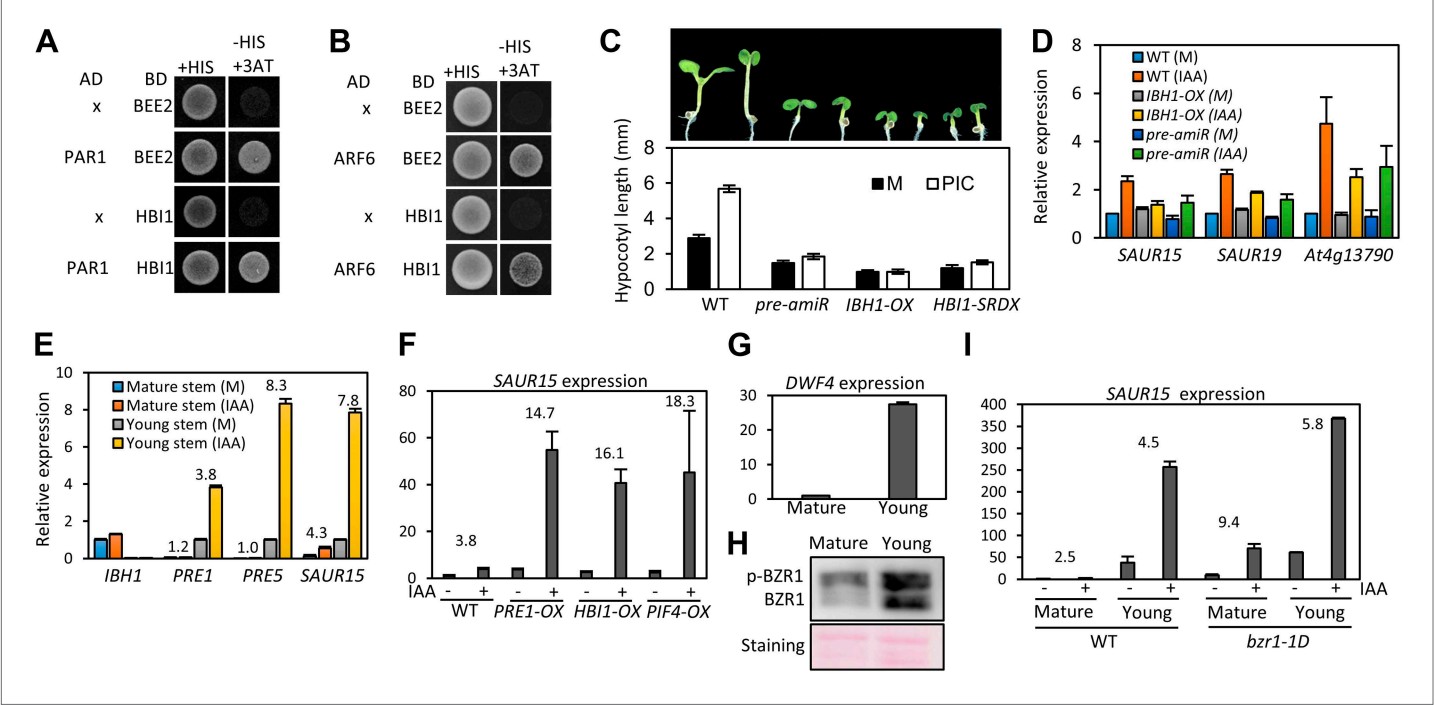

**Figure 5**. The HLH/bHLH module mediates developmental regulation of auxin sensitivity. (**A**) PAR1 interacts with BEE2 and HBI1 in yeast two-hybrid assay. (**B**) ARF6 interacts with BEE2 and HBI1 in yeast two-hybrid assay. (**C**) The *pre-amiR*, *IBH1-OX*, and *HBI1-SRDX* plants are less sensitive to auxin. Seedlings were grown on hormone-free or 5 μM picloram medium for 7 days. Error bars indicate SD (n = 10 plants). (**D**) Auxin activation of gene expression is diminished in the *pre-amiR* and *IBH1-OX* plants. 7-day-old seedlings were treated with mock (M) or 1 μM IAA for 2 hr. (**E**) Young stems are more sensitive to auxin than mature stems. Young stems (2 cm stem from the top) and mature stems (2 cm stem from the bottom) were treated with mock (M) or 1 μM IAA for 2 hr. (**F**) Auxin sensitivity of mature stem is enhanced by *PRE1-OX*, *HBI1-OX*, and *PIF4-OX*. Numbers indicate ratios between IAA-treated and mock-treated. (**G**) The *DWF4* expression is high in the young stems. (**H**) BZR1 is less phosphorylated in young stems than in mature stems. Proteins extracted from the young and mature stems of the *BZR1p::BZR1-CFP* transgenic plants were analyzed by anti-YFP immunoblotting. Ponceau S staining is shown for loading control. (**I**) Auxin sensitivity of mature stem is restored by *bzr1-1D*. Sections of mature and young stems from plants of same height were treated with IAA for 2 hr, and the expression levels of *SAUR15* were analyzed by qRT-PCR. Numbers in (**E**, **F**, **I**) indicate ratios of the expression levels of IAA-treated to mock-treated. Error bars in (**D**–**I**) indicate the SD of three biological repeats.

The following figure supplements are available for figure 5:

**Figure supplement 1**. PAR1 does not interact with ARF6 and IAA3.

RGA interacted with the middle domain and, to a lesser extent, the DNA binding domain of ARF6 (**Figure 6A**, **Figure 6—figure supplement 1**). RGA also interacted with other activator ARFs (ARF6, ARF7 and ARF8), but not repressor ARF1 (**Figure 6A**). In addition, HA-RGA, but not HA-YFP, was pulled down by ARF6-Myc in vitro (**Figure 6B**), and RGA-GFP was co-immunoprecipitated with ARF6-Myc in *Arabidopsis* protoplasts (**Figure 6C**). These results demonstrate that RGA directly interacts with ARF6. Since the middle domain of ARF6 also mediates the ARF6-PIF4/BZR1 interactions (**Figure 2—figure supplement 1**), it is likely that RGA competes with PIF4/BZR1 for interaction with ARF6. Indeed, the ARF6-PIF4 interaction was reduced by rga-Δ17, which is a stable form of RGA due to deletion of the N-terminal DELLA domain (**Bai et al., 2012b**; **Figure 6D**), suggesting that RGA disrupts the ARF6–PIF4 interaction.

To examine if RGA directly inhibits the DNA-binding ability of ARF6, like it does to BZR1 and PIF4 (**de Lucas et al., 2008**; **Feng et al., 2008**; **Li et al., 2012b**; **Gallego-Bartolome et al., 2012**; **Bai et al., 2012b**), we performed DNA–protein pull-down assays. Biotin-labeled *IAA19* promoter fragment containing AuxRE effectively pulled down MBP-ARF6 in the absence of RGA, but not after pre-incubation of MBP-ARF6 with GST-RGA (**Figure 6E**), indicating that RGA prevents ARF6 from binding to target DNA. ChIP assay confirmed that RGA inhibits ARF6 binding to target genes in vivo (**Figure 6F**). Transient reporter gene expression assays showed that ARF6 increases *IAA19* promoter activity, but

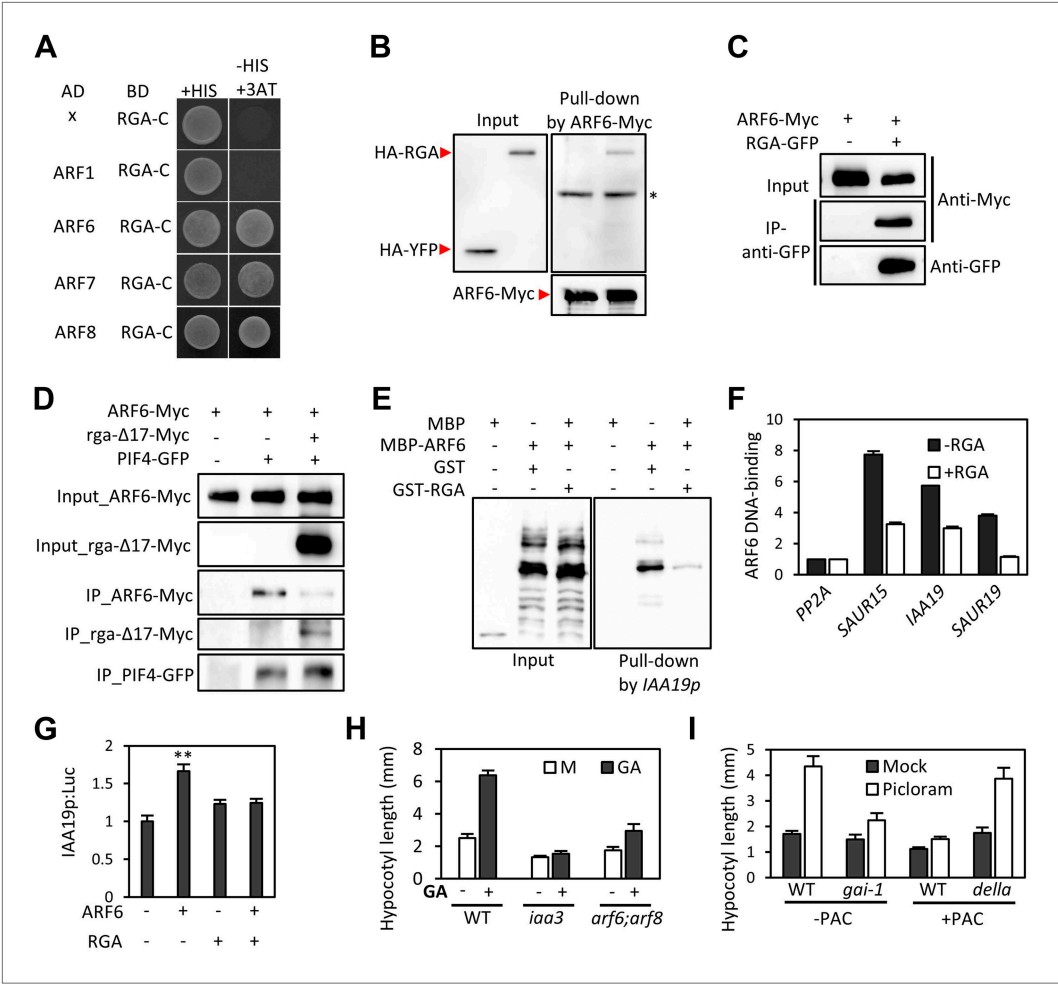

**Figure 6**. RGA interacts with ARF6 and blocks ARF6 binding to DNA. (**A**) RGA interacts with ARF6, ARF7, and ARF8, but not ARF1 in the yeast two-hybrid assay. RGA with deletion of N-terminal 208 amino acids (RGA-C) was used for the assay. (**B**) RGA interacts with ARF6 in vitro. In vitro-translated HA-YFP and HA-RGA proteins were incubated with in vitro-translated ARF6-Myc protein bound to magnetic beads, and the pulled-down proteins were analyzed by immunoblot with anti-HA antibody. * indicates IgG band. (**C**) RGA interacts with ARF6 in vivo. Protein extracts from protoplasts transfected with *ARF6-Myc* or *ARF6-Myc* and *RGA-GFP* were immunoprecipitated with anti-GFP antibody, and analyzed by immunoblots with anti-GFP or anti-Myc antibody. (**D**) RGA disrupts the PIF4–ARF6 interaction. Arabidopsis mesophyll protoplasts were transfected to express ARF6-Myc alone or with PIF4-GFP and rga-Δ17-Myc as indicated, and the extracted proteins were immunoprecipitated by anti-GFP antibody. Gel blots were probed with anti-Myc or anti-GFP antibody. (**E**) RGA inhibits ARF6 binding to the *IAA19* promoter in DNA pull-down assay. (**F**) RGA inhibits ARF6 DNA-binding ability in vivo. Protoplasts transfected with *GFP-Myc* (negative control) or *ARF6-Myc* with or without *RGA-GFP* were used for ChIP assay. Error bars indicate the s.d. of two technical repeats. Similar results were obtained in two independent experiments. (**G**) RGA inhibits ARF6 transcriptional activation activity. *IAA19p::Luc* was co-transfected with *ARF6-GFP*, *RGA-GFP*, or both, into *Arabidopsis* mesophyll protoplasts. The *IAA19p::Luc* activities were normalized by the *35S::renilla* luciferase. Error bars indicate the s.e. of 10 biological repeats ($n = 10$) and **$p < 0.01$. (**H**) Auxin signaling mutants are less sensitive to GA. Seedlings were grown on the 10 μM paclobutrazole with or without 1 μM GA in the dark. Error bars indicate SD ($n = 10$ plants). (**I**) DELLA inhibits the auxin promotion of hypocotyl elongation. Seedlings were grown on MS medium for 3 days and then transferred to the medium containing mock or 5 μM picloram, with or without 10 μM paclobutrazol (PAC), and incubated for 4 days. Error bars indicate SD ($n = 10$ plants).

The following figure supplements are available for figure 6:

**Figure supplement 1**.

**Figure supplement 2**. Auxin signaling mutants are less sensitive to GA.

the ARF6 effect was abolished by co-transfection with RGA (*Figure 6G*). These data demonstrate that DELLA interacts with ARF6 and inhibits ARF6 DNA-binding to modulate ARF6 target gene expression.

The DELLA inhibition of ARF6 suggests that GA promotes cell elongation by enhancing auxin/ARF-mediated responses. Indeed, the GA-promotion of hypocotyl elongation was much reduced in the *iaa3* and *arf6;arf8* mutants compared to the wild type (*Figure 6H*, *Figure 6—figure supplement 2*), supporting the notion that auxin activation of ARFs is necessary for the GA promotion of hypocotyl elongation. In addition, the auxin promotion of hypocotyl elongation was compromised in the GA-insensitive mutant *gai-1* (*Peng et al., 1997*) and in wild-type plants grown on the medium containing GA biosynthesis inhibitor paclobutrazol (PAC) (*Figure 6I*), both of which accumulate DELLA proteins, but was nearly normal in the PAC-treated *della pentuple* mutant (*della*) lacking all five members of the DELLA family (*Wang et al., 2009*; *Figure 6I*), indicating that accumulation of DELLA proteins inhibits auxin sensitivity. Taken together, these data indicate that the GA-induced degradation of DELLAs allows ARF6, together with BZR1 and PIF4, to bind to target DNA and to activate gene expression and hypocotyl cell elongation.

## Discussion

The ability to make decision based on integration of large numbers of signal inputs is a feature of advanced control system. The level of such ability that has evolved for the cellular control systems of plants remains unclear. While cell elongation, as the major growth process in plants, is the target of many signaling pathways that have evolved to provide the high level of developmental plasticity in higher plants, it has been unclear whether these pathways act on independent cellular machineries involved in elongation or they are processed by a central control system into a coherent cellular decision. Although physiological synergism and genetic interdependency suggested signal integration, the essential molecular connections between the signaling pathways, particularly with the auxin pathway, have been elusive. Our findings of the BZR-ARF-PIF/DELLA (BAP/D) module illustrate an elegant model of signal integration, which explains the synergistic or antagonistic interactions between auxin and the other signaling pathways. Our study demonstrates that hypocotyl cell elongation is controlled by major hormonal and environmental pathways through a central circuit of interacting transcription regulators.

A close relationship between auxin and BR has been suggested by their synergistic physiological effects, genetic interdependence, and overlapping genomic effects (*Goda et al., 2004*; *Nemhauser et al., 2004*). Several possible mechanisms have been proposed for the auxin-BR interdependence, including BIN2–ARF interaction, and co-regulation of gene expression by BZR and ARF factors (*Vert et al., 2008*; *Walcher and Nemhauser, 2012*). Our genetic analysis demonstrated that BZR1 plays a major role in potentiating auxin responses. Consistent with our genetic data and previous analysis of BR-auxin co-regulation of the *SAUR15* promoter (*Walcher and Nemhauser, 2012*), our genomic and biochemical experiments showed direct interaction between BZR1 and ARF6 at the promoters of a large set of genomic targets, demonstrating that the auxin and BR pathways mainly converge through BZR–ARF interaction at shared target promoters.

Light antagonizes auxin, BR, and GA to inhibit hypocotyl elongation and promote photomorphogenesis. Light signaling mediated by phytochromes induces degradation of PIFs (*Leivar and Quail, 2011*). In this study, we show that ARF6 shares most of its binding target genes with PIF4 and they interdependently activate shared target promoters, which explains the requirement of PIFs for auxin responsive gene expression and hypocotyl elongation and the requirement of auxin for skotomorphogenesis. The incomplete overlap between targets of BZR1, ARF6, and PIF4 suggests that their interactions are selected and/or facilitated by specific promoters to allow co-regulation of hypocotyl elongation and photomorphogenesis, while each pathway or combination of two pathways may regulate other gene sets and developmental processes. ARF6 showed a higher level of target overlap with PIF4 than with BZR1, suggesting a tighter integration between the environmental signal and hormonal signal than between different hormones.

PIFs are considered a central hub for integrating environmental and developmental signals (*Leivar and Quail, 2011*). In addition to light, temperature and the circadian clock transcriptionally regulate members of PIF family (*Nozue et al., 2007*; *Koini et al., 2009*), and consequently alter auxin synthesis as well as auxin sensitivity through the BAP module. The integral role of PIFs in hormone-responsive gene expression is also consistent with the finding of the HUD element, a potential binding site of PIF4/5, associated with the morning-specific phytohormone gene expression program (*Nozue et al.,*

*2007*; *Michael et al., 2008*). The DNA-binding activities of PIFs are also modulated by the tripartite HLH/bHLH module, in which the PRE family of non-DNA binding HLH factors sequesters another class of HLH factors (including IBH1 and PAR1), which otherwise inhibit DNA-binding of bHLH factors including HBI1 and PIFs (*Hao et al., 2012*; *Ikeda et al., 2012*; *Bai et al., 2012a*). The HLH/bHLH module has a major effect on plant sensitivity to auxin, BR and GA, presumably by controlling PIF4 and HBI1, which both interact with ARF6. Considering that GA, BR, auxin, and PIF4 increase the transcription levels of several *PRE* members (*Chapman et al., 2012*; *Oh et al., 2012*; *Bai et al., 2012b*), the HLH/bHLH modules appear to form positive feedback loops, which potentially re-enforce the activation of the BAP module and help maintain the growing condition in the dark or in young developing organs. The increase of *IBH1* expression appears to mediate, at least partly, inactivation of hormone responses in mature organs (*Figure 5*), whereas a decrease of *HBI1* expression mediates growth arrest and defense activation in response to pathogen infection (*Fan et al., 2014*; *Malinovsky et al., 2014*). As such, the HLH/bHLH module also provides additional nodes for input and output.

GA has been shown to promote hypocotyl elongation by removing DELLA repression of BZR1 and PIF4. Physiological studies supported additive effects of auxin and GA, while a recent study suggested that auxin regulates GA biosynthesis to release DELLA-dependent growth repression (*Chapman et al., 2012*). In this study, we show that GA promotion of hypocotyl elongation also requires auxin activation of ARFs, as the *iaa3* and *arf6,arf8* mutants show severely reduced GA response in hypocotyl elongation. Similar to DELLA inhibition of BZR1 and PIFs, DELLA also inhibits DNA-binding of ARF6. In contrast to the cooperative interactions among BZR1, ARF6, and PIF4, the DELLA protein RGA inhibits ARF6–PIF4 interaction. As such, DELLA inhibits both protein–DNA and protein–protein interactions of the BAP module, providing presumably coordinated and coherent control of all three components of the BAP module. Together, our study illustrates that the major growth-regulation pathways, auxin, BR, GA, and phytochrome, converge at the BAP/D module to control hypocotyl cell elongation (*Figure 7*). We propose that the BAP/D module coupled with the HLH/bHLH module forms the central growth regulation network that integrates hormonal, environmental, and developmental inputs into the decisions about hypocotyl cell elongation (*Figure 7*).

In different developmental contexts, auxin induces distinct cellular and developmental responses, such as cell division, differentiation, elongation, and organogenesis. Whether the BAP/D network or similar transcription factor modules contribute to other auxin signaling outputs remains to be elucidated by future studies. The lack of interaction of BZR1 and PIF4 with ARF7 or ARF1 indicates a level of specificity of signal integration through ARF6 and ARF8 for regulation of hypocotyls and other shoot organs where ARF6 and ARF8 play important roles. Recent studies showed that auxin also acts through a receptor kinase-mediated signaling pathway to regulate cell morphogenesis (*Xu et al., 2014*); the details of this new auxin pathway and its relationships with the ARF-mediated auxin pathway and other hormonal and environmental signaling pathways are yet to be elucidated by future studies. Apparently, hormone interactions are complex and vary with developmental context. While the BAP/D module explains how auxin, BR, GA, light, and temperature coordinately regulate cell elongation of the hypocotyl, and likely other shoot organs in *Arabidopsis*, mechanisms of signal integration in other developmental contexts might be different and thus remain be elucidated.

## Materials and methods

### Plant materials and growth conditions

All the *Arabidopsis thaliana* plants used in this study were in Col-0 ecotype background, except *gai-1* and *della*, which were in the Landsberg *erecta* ecotype background (*Wang et al., 2009*). The *arf6;arf8* double mutant (*arf6-2;arf8-3*) was provided by Jason W Reed (*Nagpal et al., 2005*). To generate *ARF6p::ARF6-Myc* transgenic lines, *ARF6* genomic fragment including 2.5 kb upstream of transcription start was cloned into the gateway compatible *p1390-4Myc-His* vector and transformed into the Col-0 and *arf6(−/−);arf8(+/−)*.

### Co-immunoprecipitation (co-IP) assays

Transgenic plants expressing ARF6-Myc and BZR1-YFP from 35S promoter (*ARF6-Myc;BZR1-YFP*) or ARF6-Myc and PIF4-YFP from 35S promoter (*ARF6-Myc;PIF4-YFP*) were treated with 100 nM BL for 4 hr. Harvested tissues were grounded in liquid nitrogen, homogenized in IP buffer (50 mM Tris-Cl pH7.5, 1 mM EDTA, 75 mM NaCl, 0.1% Triton X-100, 5% Glycerol, 1 mM PMSF, 1x Protease Inhibitor). After centrifugation at 20,000×*g* for 10 min, 1 ml of supernatant was incubated for 1 hr with anti-GFP

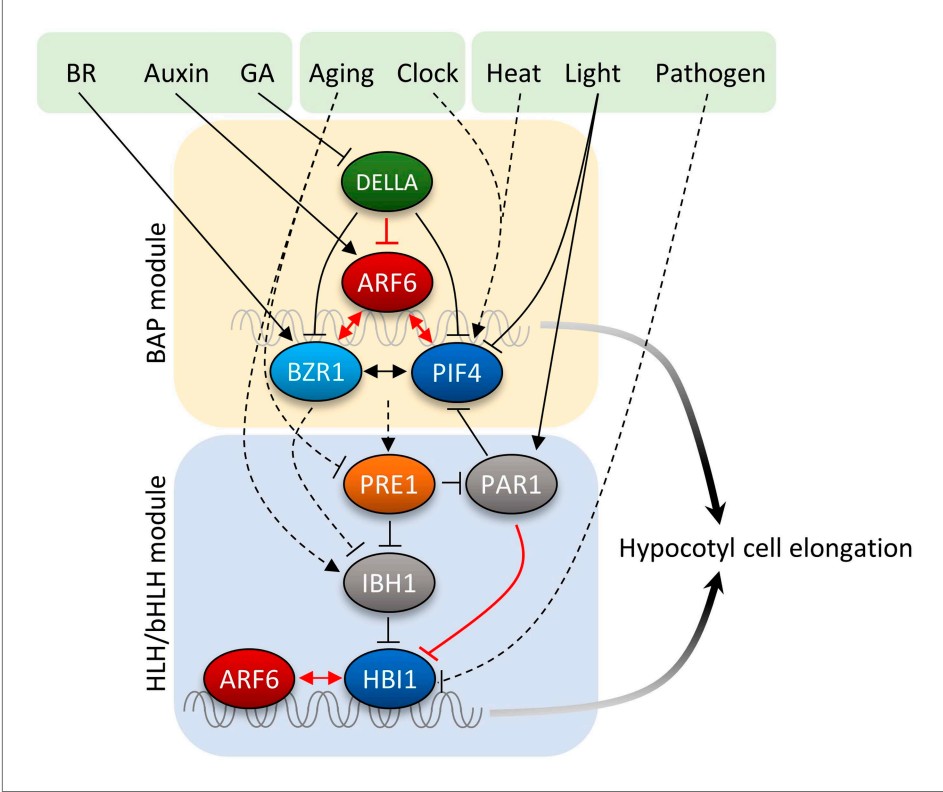

**Figure 7**. Diagram of the central growth regulation circuit. In the diagram, solid lines indicate protein–protein interaction or post-translational modification, and dashed lines indicate transcriptional regulation. Red lines indicate new discoveries made in this study. In the BAP module, all three transcription factors, BR-regulated BZR1, auxin-regulated ARF6, and light/temperature-regulated PIF4, interact with each other and cooperatively regulate shared target genes and hypocotyl cell elongation. GA-regulated DELLA interacts with all BAP transcription factors and inhibits their DNA binding. Downstream of BAP module, the HLH/bHLH module, consisting of PRE1, IBH1/ PAR1 and HBI1/PIF, modulates BAP activities through HLH–bHLH interactions. The BAP transcription factors positively regulate PRE1 in the HLH/bHLH module forming positive feedback loops. Development and pathogen signals are integrated into the central growth regulation network through PRE1/IBH1 and HBI1, respectively.

(custom made, 5 µg) immobilized on protein A/G agarose beads (Pierce Biotechnology, Rockford, IL). The beads were then washed for three times with 1 ml of IP buffer and eluted samples were analyzed by immunoblot using anti-Myc (Cell Signaling Technology, Beverly, MA) and anti-GFP antibodies.

For co-IP assays using *Arabidopsis* mesophyll protoplasts, $2 \times 10^4$ isolated mesophyll protoplast were transfected with a total 20 µg of DNA and incubated overnight. Total proteins were extracted from the protoplasts using the IP buffer, and immunoprecipitation was performed as described above.

## Transient gene expression assays

Isolated *Arabidopsis* mesophyll protoplasts ($2 \times 10^4$) were transfected with a total 20 µg of DNA and incubated overnight. Protoplasts were harvested by centrifugation and lysed in 50 µl of passive lysis buffer (Promega, Madison, WI). Firefly and Renilla (as an internal standard) luciferase activities were measured by using a dual-luciferase reporter kit (Promega).

## qRT-PCR gene expressions analysis

Total RNA was extracted from seedlings treated with mock or specific hormones by using the Spectrum Plant Total RNA kit (Sigma, St. Louis, MO). M-MLV reverse transcriptase (Fermentas, Thermo Scientific, Waltham, MA) was used to synthesize cDNA from the RNA. Quantitative real-time PCR (qRT-PCR) was performed using LightCycler 480 (Roche, Basel, Switzerland) and the Bioline SYBR green master mix (Bioline). Gene expression levels were normalized to that of PP2A and are shown relative to the expression levels in wild type. Gene specific primers are listed in *Supplementary file 1*.

## Chromatin immunoprecipitation (ChIP) assays

For ChIP assays, seedlings (*35S::ARF6-Myc*) were cross-linked for 20 min in 1% formaldehyde under vacuum. The chromatin complex was isolated, resuspended in lysis buffer (50 mM Tris–HCl pH 8.0, 10 mM EDTA, 200 mM NaCl, 0.5% Triton X-100, 1 mM PMSF) and sheared by sonication to reduce the average DNA fragment size to around 500 bps. The sonicated chromatin complex was immunoprecipitated by anti-Myc antibody (Cell Signaling Technology)-bound protein A agarose beads (Millipore, Bedford, MA). The beads were washed with low-salt buffer (50 mM Tris–HCl at pH 8.0, 2 mM EDTA, 150 mM NaCl, 0.5% Triton X-100), high-salt buffer (50 mM Tris–HCl at pH 8.0, 2 mM EDTA, 500 mM NaCl, 0.5% Triton X-100), LiCl buffer (10 mM Tris–HCl at pH 8.0, 1 mM EDTA, 0.25 M LiCl, 0.5% NP-40, 0.5% deoxycholate), and TE buffer (10 mM Tris–HCl at pH 8.0, 1 mM EDTA) and eluted with elution buffer (1% SDS, 0.1 M NaHCO3). The ARF6-bound DNA was purified by using a PCR purification kit (Thermo Scientific) and analyzed by ChIP-qPCR. The enrichment of DNA was calculated as the ratio between ARF6-Myc and WT samples, normalized to that of the *PP2A*. Primers for qPCR are listed in *Supplementary file 1*.

## ChIP-Seq analysis

The 5-day-old dark-grown seedlings of *ARF6p::ARF6-Myc;arf6;arf8* and *35S::GFP-Myc* (control), or *BZR1p::BZR1-CFP* and *35S::YFP* (control) were used for ARF6 or BZR1 ChIP-Seq analysis, respectively, following protocols described before (*Oh et al., 2012*). For ChIP-Seq library construction, 10 ng of ChIP-DNA were pooled from three biological repeats to reduce sample variation. High-throughput sequencing of ChIP-Seq libraries was carried out on an Illumina HiSeq 2000. Sequences in Solexa FASTQ format were mapped to the *Arabidopsis* genome, TAIR9, using SOAP2. ARF6 binding peaks were identified using ChIP-Seq analysis R (CSAR) software with parameters (backg = 10, norm = −1, test = 'Ratio', times = 1e6, digits = 2) (*Muino et al., 2011*). Binding peaks with FDR < 0.01 were defined as the ARF6 binding peak and used in further analyses. Genes having at least one ARF6 binding peak within its promoter (−3 kb), coding region or 1 kb downstream from stop codon were considered ARF6 binding target genes.

## ChIP-reChIP assays

ChIP-reChIP assays were performed using anti-Myc antibody first (Cell Signaling Technology) and then using anti-GFP antibody (custom made). Precipitated DNA was quantified by qPCR. Enrichment of DNA was calculated as the ratio between transgenic plants and wild type control, normalized to that of the *PP2A* coding region as an internal control. All error bars indicate the SD of three biological repeats.

## RNA-Seq analysis

Seedlings were grown on medium containing 2 μM propiconazole (PPZ) in the dark for 5 days and treated with mock or 100 nM BL for 4 hr before harvesting. Total RNA was extracted by using the Spectrum Plant Total RNA kit (Sigma). Libraries were constructed by using TruSeq RNA Sample Preparation Kit (Illumina) according to the manufacturer's instruction. RNA-Seq analysis was performed as described previously (*Oh et al., 2012*). Differentially expressed genes were defined by a 1.5-fold difference between samples with p<0.01.

## Protein pull-down assays

The ARF6-Myc, HA-RGA and HA-YFP proteins were synthesized by TNT T7 Quick Coupled in vitro transcription/translation system (Promega). The ARF6-Myc proteins were pre-incubated with anti-Myc antibody (Cell Signaling Technology)-bound protein A-Dynabeads (Life Technology) for 2 hr. After removing unbound ARF6-Myc proteins, the HA-RGA or HA-YFP proteins were incubated with the ARF6-Myc-bound Dynabeads for 1 hr in PBSN buffer (PBS buffer + 0.1% NP-40). The beads were washed three times with the PBSN buffer and the pulled-down proteins were analyzed by immunoblots using anti-HA antibody (Roche) and anti-Myc antibody (Cell Signaling Technology).

## DNA pull-down assays

The MBP and MBP-ARF6 proteins were affinity-purified from *Escherichia coli* by using amylose resin (NEB). The *IAA19* promoter fragment was amplified by PCR using biotin-labeled primers (*Supplementary file 1*). The biotin-labeled *IAA19* promoter fragment and the MBP or MBP-ARF6 proteins were incubated with streptavidin-bound agarose beads (Sigma) for 1 hr in IP100 buffer (100 mM potassium glutamate, 50 mM Tris–HCl pH 7.6, 2 mM MgCl$_2$, 0.05% NP-40). The beads were washed four times with the IP100 buffer and DNA-bound proteins were analyzed by an immunoblot.

### GEO accession numbers
The ChIP-seq data used in this study may be viewed under GSE51770.
The RNA-seq data used in this study may be viewed under GSE51772.

## Acknowledgements
We thank Jason W Reed for providing seeds of the *arf6;arf8* double mutant and Dr Ulrich Kutschera for comment on the manuscript. Research was supported by a grant from the National Institutes of Health (NIH R01GM066258).

## Additional information

### Funding

| Funder | Grant reference number | Author |
| --- | --- | --- |
| HHS\|NIH\|National Institute of General Medical Sciences (NIGMS) | R01GM066258 | Jia-Ying Zhu, Ming-Yi Bai, Yu Sun, Zhi-Yong Wang |

The funders had no role in study design, data collection and interpretation, or the decision to submit the work for publication.

### Author contributions
EO, Conception and design, Acquisition of data, Analysis and interpretation of data, Drafting or revising the article; J-YZ, M-YB, Conception and design, Acquisition of data; RAA, YS, Acquisition of data, Contributed unpublished essential data or reagents; Z-YW, Conception and design, Analysis and interpretation of data, Drafting or revising the article

## Additional files

### Supplementary file
• Supplementary file 1. Primer list for qRT-PCR, ChIP-PCR and DNA pull-down assays.

### Major datasets

The following datasets were generated:

| Author(s) | Year | Dataset title | Dataset ID and/or URL | Database, license, and accessibility information |
| --- | --- | --- | --- | --- |
| Oh E, Wang Z | 2014 | Interactions of ARF6 with PIF4, BZR1, and RGA integrate auxin signaling with environmental and other hormonal signals in Arabidopsis [ChIP-Seq] | GSE51770; http://www.ncbi.nlm.nih.gov/geo/query/acc.cgi?acc=GSE51770 | Publicly available at the Gene Expression Omnibus (http://www.ncbi.nlm.nih.gov/geo/) |
| Oh E, Wang Z | 2014 | Interactions of ARF6 with PIF4, BZR1, and RGA integrate auxin signaling with environmental and other hormonal signals in Arabidopsis [RNA-Seq] | GSE51772; http://www.ncbi.nlm.nih.gov/geo/query/acc.cgi?acc=GSE51772 | Publicly available at the Gene Expression Omnibus (http://www.ncbi.nlm.nih.gov/geo/) |

The following previously published datasets were used:

| Author(s) | Year | Dataset title | Dataset ID and/or URL | Database, license, and accessibility information |
| --- | --- | --- | --- | --- |
| Oh E, Wang Z | 2012 | Interaction between BZR1 and PIF4 integrates brassinosteroid and environmental responses [ChIP-seq] | GSE35315; http://www.ncbi.nlm.nih.gov/geo/query/acc.cgi?acc=GSE35315 | Publicly available at the Gene Expression Omnibus (http://www.ncbi.nlm.nih.gov/geo/) |

| Oh E, Wang Z | 2012 | Interaction between BZR1 and PIF4 integrates brassinosteroid and environmental responses | GSE37160; http://www.ncbi.nlm.nih.gov/geo/query/acc.cgi?acc=GSE37160 | Publicly available at the Gene Expression Omnibus (http://www.ncbi.nlm.nih.gov/geo/) |

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
