## [Decision Letter]

Thank you for sending your work entitled “Cell elongation is regulated through a central circuit of interacting transcription factors in Arabidopsis hypocotyl” for consideration at *eLife*. Your article has been favorably evaluated by Detlef Weigel (Senior editor) and 3 reviewers, one of whom, Sheila McCormick, is a member of our Board of Reviewing Editors, and one of whom, Enamul Huq, has agreed to reveal his identity.

The Reviewing editor and the other reviewers discussed their comments before we reached this decision, and the Reviewing editor has assembled the following comments to help you prepare a revised submission.

You have previously reported how light and BR signaling converge to regulate hypocotyl elongation. Here you provide substantial evidence to add auxin to the mix, and in addition you demonstrate how GA signaling inhibitors, DELLA proteins, repress these signals by directly interacting with ARF6, BZR1, and PIF4. The manuscript is very comprehensive, the data are largely convincing, and the in vitro interactions are confirmed by genetic analysis and/or ChIP experiments. Your manuscript therefore adds substantially to our knowledge of how these diverse growth pathways interact. We have listed below the major points that must be addressed in the revision.

1) It is important to address whether ARF8 also interacts with PIF4 and BZR1, as ARF6 and 8 are redundant in hypocotyl elongation; Y2H or other in vitro data would be sufficient. Additionally, the reasons why ARF6 was chosen should be more explicitly explained.

2) Does the same ARF domain mediate interactions with RGA and BZR1 and/or PIF4, and does RGA disrupt the ARF6-BZR1 or ARF6-PIF4 interaction? Is this a competitive interaction?

3) In Figure 4, you compared the phenotypes of iaa3 mutants to the arf6,8 mutants. Although the generic model posits that IAA proteins inhibit ARF function, the effect of iaa3 can't be explained as a defect in arf6/8. You should cite a reference or establish that IAA3 functions through ARF6/8.

4) In Figure 5 and the Results section, you conclude that BZR1 is more active in young tissues. However, there is no direct evidence that the activity of BZR1 is enhanced. In fact, the protein level is higher in young tissues as opposed to its activity. This contention needs experimental support.

5) The quality of Figure 6 needs improvement, as it is and not very convincing support for the RGA-ARF6 interaction. Figure 6 requires genetic evidence, such as analysis of della mutants, to support the model. The conclusions are solely based on GA treatment.

6) In the Discussion section, you conclude that the BIN2-ARF connection is less important, but this statement is premature given that the phenotypes were not compared in the same assay. This should be addressed experimentally or the statement must be tempered.

Although not required, it would be conceptually interesting if you can provide information as to how ARF6 interacts with both PIF4 and BZR1, e.g., do the proteins form a trimeric complex?

---

## [Author Response]

*1) It is important to address whether ARF8 also interacts with PIF4 and BZR1, as ARF6 and 8 are redundant in hypocotyl elongation; Y2H or other in vitro data would be sufficient. Additionally, the reasons why ARF6 was chosen should be more explicitly explained*.

We have conducted a yeast two-hybrid assay with ARF8. The results clearly show that ARF8 interacts with both PIF4 and BZR1 (Figure 2—figure supplement 2). We have added a sentence at the beginning of Results to clearly explain why we selected ARF6 for further analyses as follows.

“ARF6 and its closed homolog ARF8 were previously shown to redundantly regulate hypocotyl elongation in *Arabidopsis* (31).”

*2) Does the same ARF domain mediate interactions with RGA and BZR1 and/or PIF4*, *and does RGA disrupt the ARF6-BZR1 or ARF6-PIF4 interaction? Is this a competitive interaction?*

To address this question, we have first performed yeast two-hybrid assays. The results show that the middle domain and, to a lesser extent, the DNA binding domain of ARF6 mediate the ARF6-RGA interaction (Figure 6—figure supplement 1). Since the middle domain of ARF6 is required for the ARF6-PIF4 and ARF6-BZR1 interactions, it is likely that RGA and PIF4/BZR1 competitively interact with ARF6. We then performed co-immunoprecipitation assays using mesophyll protoplasts. As shown in Figure 6—figure supplement 2, the ARF6-PIF4 interaction was reduced by co-transfection with rga (stabilized RGA due to lack of DELLA domain) indicating that RGA disrupts the ARF6-PIF4 interaction.

*3) In*
Figure 4*, you compared the phenotypes of iaa3 mutants to the arf6/8 mutants. Although the generic model posits that IAA proteins inhibit ARF function, the effect of iaa3 can't be explained as a defect in arf6/8. You should cite a reference or establish that IAA3 functions through ARF6/8*.

AUX/IAA proteins like IAA3 inhibit ARF functions through directly interacting with ARFs. To show activities of ARF6 and ARF8 are regulated by IAA3, we have performed yeast two-hybrid assays. The results show that IAA3 interacts with both ARF6 and ARF8 (Figure 3—figure supplement 1). We also cited a paper (45) reporting IAA3-ARF6 and IAA3-ARF8 interactions.

*4) In*
Figure 5
*and the Results section, you conclude that BZR1 is more active in young tissues. However, there is no direct evidence that the activity of BZR1 is enhanced. In fact, the protein level is higher in young tissues as opposed to its activity. This contention needs experimental support*.

The phosphorylation of BZR1 have been shown to regulate BZR1 activity in many ways including destabilizing BZR1 protein and inhibiting BZR1 nuclear localization and DNA binding (20, 38, 17). Thus, BZR1 activity can be inferred from its phosphorylation status. In Figure 5, BZR1 accumulated more and is less phosphorylated in the young stem than in the mature stem. The result suggests that BZR1 is more activated in the young stem. We have cited these papers.

*5) The quality of*
Figure 6
*needs improvement, as it is and not very convincing support for the RGA-ARF6 interaction*.

As the reviewers’ suggestion, we performed the in vitro pull down assays again with TNT in vitro-translated proteins. The result clearly shows that RGA, but not YFP alone, interacts with ARF6 (new Figure 6).

Figure 6
*requires genetic evidence, such as analysis of della mutants, to support the model. The conclusions are solely based on GA treatment*.

To further support the function of ARF-DELLA interaction in regulating the hypocotyl elongation, as reviewers’ suggestion, we tested auxin responses of GA-insensitive mutant (*gai-1*) and *della pentuple* mutant (*della*), and also examined an effect of GA biosynthesis inhibitor paclobutrazol (PAC) on the auxin response. The auxin response (hypocotyl elongation) was compromised in *gai-1* and on the PAC medium (high DELLA condition), but the reduced auxin sensitivity on the PAC medium was restored in *della* (low DELLA condition) (Figure 6) indicating that DELLA inhibits the auxin promotion of hypocotyl elongation. We believe these genetic data further support our model of DELLA inhibiting ARF activities.

*6) In the Discussion section, you conclude that the BIN2-ARF connection is less important, but this statement is premature given that the phenotypes were not compared in the same assay. This should be addressed experimentally or the statement must be tempered*.

We agree and deleted the sentence.

*Although not required, it would be conceptually interesting if you can provide information as to how ARF6 interacts with both PIF4 and BZR1, e.g.*, *do the proteins form a trimeric complex?*

To address this question, we examined if the ARF6-PIF4 interaction is affected by BZR1. If BZR1 and PIF4 compete with each other for interaction with ARF6 (no trimeric complex), BZR1 would disrupt the ARF6-PIF4 interaction. On the other hand, if BZR1, PIF4 and ARF6 form the trimeric (or bigger) complex, BZR1 would either enhance the ARF6-PIF4 interaction or have no effect on the ARF6-PIF4 interaction. As shown in Figure 2, the ARF6-PIF4 interaction is increased by bzr1-1D, indicating that BZR1 enhances the ARF6-PIF4 interaction, most likely through trimeric interaction.